# CT-Free Attenuation Correction in Paediatric Long Axial Field-of-View Positron Emission Tomography Using Synthetic CT from Emission Data

**DOI:** 10.3390/diagnostics14242788

**Published:** 2024-12-12

**Authors:** Maria Elkjær Montgomery, Flemming Littrup Andersen, René Mathiasen, Lise Borgwardt, Kim Francis Andersen, Claes Nøhr Ladefoged

**Affiliations:** 1Department of Clinical Physiology and Nuclear Medicine, Copenhagen University Hospital, Rigshospitalet, 2100 Copenhagen, Denmark; maria.elkjaer.montgomery@regionh.dk (M.E.M.); flemming.andersen@regionh.dk (F.L.A.); lise.borgwardt@regionh.dk (L.B.); kim.francis.andersen.01@regionh.dk (K.F.A.); 2Department of Clinical Medicine, University of Copenhagen, 2200 Copenhagen, Denmark; rene.mathiasen@regionh.dk; 3Department of Paediatrics and Adolescent Medicine, Copenhagen University Hospital, Rigshospitalet, 2100 Copenhagen, Denmark; 4Department of Applied Mathematics and Computer Science, Technical University of Denmark, 2800 Kgs. Lyngby, Denmark

**Keywords:** paediatric, oncology, PET/CT, synthetic CT, deep learning, artificial intelligence

## Abstract

**Background/Objectives**: Paediatric PET/CT imaging is crucial in oncology but poses significant radiation risks due to children’s higher radiosensitivity and longer post-exposure life expectancy. This study aims to minimize radiation exposure by generating synthetic CT (sCT) images from emission PET data, eliminating the need for attenuation correction (AC) CT scans in paediatric patients. **Methods**: We utilized a cohort of 128 paediatric patients, resulting in 195 paired PET and CT images. Data were acquired using Siemens Biograph Vision 600 and Long Axial Field-of-View (LAFOV) Siemens Vision Quadra PET/CT scanners. A 3D parameter transferred conditional GAN (PT-cGAN) architecture, pre-trained on adult data, was adapted and trained on the paediatric cohort. The model’s performance was evaluated qualitatively by a nuclear medicine specialist and quantitatively by comparing sCT-derived PET (sPET) with standard PET images. **Results**: The model demonstrated high qualitative and quantitative performance. Visual inspection showed no significant (19/23) or minor clinically insignificant (4/23) differences in image quality between PET and sPET. Quantitative analysis revealed a mean SUV relative difference of −2.6 ± 5.8% across organs, with a high agreement in lesion overlap (Dice coefficient of 0.92 ± 0.08). The model also performed robustly in low-count settings, maintaining performance with reduced acquisition times. **Conclusions**: The proposed method effectively reduces radiation exposure in paediatric PET/CT imaging by eliminating the need for AC CT scans. It maintains high diagnostic accuracy and minimises motion-induced artifacts, making it a valuable alternative for clinical application. Further testing in clinical settings is warranted to confirm these findings and enhance patient safety.

## 1. Introduction

PET/CT imaging plays a key role in many fields of medicine, notably in oncology. While its application is crucial for both adults and children, PET/CT in paediatric imaging comes with added challenges, as children are at a higher risk of the adverse effects of radiation compared to adults [1]. This is both because children’s tissue and organs are more radiosensitive as they are not fully mature and because children have a higher postexposure life expectancy where the negative effects of radiation can manifest [2,3,4]. When optimising PET/CT imaging for paediatrics it is therefore particularly relevant to focus on minimising radiation. Possible strategies to achieve this include eliminating unnecessary diagnostic examinations, decreasing the tracer dose needed for PET imaging [5,6], and reducing the CT dose [7].

In PET, artificial intelligence (AI) has been proposed to remove the noise associated with reducing the radiation dose [8]. For paediatric imaging, the focus has primarily been on PET/MRI as the scanner allows for CT-free examination of the patients, thus removing the dose exposure from CT altogether. Theruvath et al. demonstrated retained clinical quality with a 50% dose reduction in whole-body PET examinations of children and young adults with lymphoma using a commercially available method [9]. Wang et al. proposed a convolutional neural network (CNN) trained with an attention mask that allowed for a reduction of dose to one-sixteenth of the original dose when given both the ultra-low-dose PET and MRI as input. The authors showed retained clinical accuracy and lesion quantifiability in a cohort of 23 paediatric patients with lymphoma [10].

With the advent of Long Axial Field-of-View (LAFOV) PET/CT scanners, such as the Biograph Vision Quadra, Siemens Healthineers [11], and the uEXPLORER, United Imaging [12], it is now possible to reduce the administered radiotracer activity by a factor of ten or more without loss of diagnostic quality primarily due to the larger detector coverage compared to traditional PET/CT systems [13]. These findings have been reproduced in paediatric cohorts [14,15,16]. Therefore, especially in combination with low-dose AI-based methods, the main contributor of radiation dose in a PET/CT examination is now the attenuation correction (AC) CT required for accurate quantification of PET images in the PET/CT setup. To further reduce the amount of ionising radiation that the patient is exposed to, it thus makes sense to focus on radiation minimisation associated with the CT scan.

One approach is to synthesise high-quality CT images from lower-dose CT images [17]. Several studies have achieved this using deep learning architectures such as CNN [18,19,20] and generative adversarial networks (GAN) [21,22]. Initially, these studies were mostly focused on adults, but more recently, studies focusing on paediatric patients have also been proposed [23,24,25,26,27,28,29].

An alternative strategy involves the elimination of the CT scan altogether. One approach is to use a separate MRI for attenuation correction [30]. MRI-based attenuation correction poses a challenge with paediatric patients using traditional segmentation- or atlas-based methods [31,32]. While AI-based MRI-CT methods have been proposed [33,34,35], only a few of these have been applied to paediatric patients [36], in part due to the scarcity of paired training data. Furthermore, MRI is not always available for LAFOV PET imaging, and when it is, there is the challenge of registration due to the movement of the patient during scanning [37].

Finally, another approach is therefore a PET-derived solution for attenuation correction. Here, the non-attenuation corrected (NAC) PET can be used to synthesise the attenuation and scatter-corrected PET image directly or to synthesise the AC CT image. While the direct NAC PET-to-PET methods have demonstrated promising results [38,39,40,41], they come with the disadvantage that they are hard to debug when determining whether errors such as hallucinations have occurred during synthesis. Contrary to this, the NAC PET-to-CT methods could better fit the clinical workflow as artifacts in the synthetic CT are easily identifiable, making it easier to visually evaluate the output of these models. Several studies have achieved this with positive results [42,43,44,45,46,47,48]. These studies all include datasets from adult patients, but as radiation minimisation is especially relevant for paediatric patients, a model designed specifically for children would be of value.

Inspired by the studies above, the purpose of this project is to generate synthetic CT images from NAC PET images from paediatric patients scanned on the LAFOV Biograph Vision Quadra PET/CT, thereby eliminating the need for an AC CT acquisition. The developed model will make use of our previously proposed model for the adult cohort [47]. We furthermore explore the need for separate models on lower count-rates to accommodate the research towards reducing scan time or PET radiotracer dose.

## 2. Materials and Methods

### 2.1. Patient Cohort

The cohort consisted of 195 paired NAC PET and CT images from 128 consecutively included paediatric patients injected with [^18^F]FDG acquired from August 2021 to June 2023 from Rigshospitalet (Copenhagen, Denmark). From this cohort, 81 of the images were obtained on a Siemens Biograph Vision 600 PET/CT, and the remaining 114 were obtained from a LAFOV Siemens Vision Quadra PET/CT scanner, Siemens Healthineers, Knoxville, TN, USA. All patients examined with the Vision scanner were used for training, while the patients examined on the Quadra scanner were split into training and test sets. We used stratified sampling, where we first divided the patients based on age and weight (<6 years old or <40 kg) and sampled 25% in each group for the test set. The split was performed on the subject level, ensuring repeat scans of a patient were in the same set. The patient information is shown in Table 1.

### 2.2. Data Acquisition and Pre-Processing

The dataset included in this study consists of patients from either a Siemens Biograph Vision 600 or a LAFOV Siemens Vision Quadra PET/CT scanner. An ultra-low-dose CT (ULDCT) was acquired (ref mAs 7.0). All scans are without IV contrast. [^18^F]FDG images were acquired according to the European guidelines [49], with [^18^F]FDG administered at 1.5 MBq/kg body weight (Quadra data) or 3 MBq/kg body weight (Vision data) 60 min prior to scanning for 5 min (*n* = 19 Quadra) or 10 min (all Vision, *n* = 95 Quadra). The dataset includes patients in both arms up and arms down positioning.

All CT images have a matrix size of 512 × 512 and a pixel dimension of 1.52 × 1.52 × 2 mm^3^. PET data were reconstructed using e7tools (Siemens Healthineers, Knoxville, TN, USA), with parameters identical to the clinical setting. For the purpose of this study, we reconstructed the NAC PET images without attenuation correction using 3D ordinary Poisson OSEM (3D-OP-OSEM), four iterations and five subsets, and point spread modelling (PSF) applied. Post-filtering was set at 2 mm (Quadra data) or 4 mm (Vision data). All PET images had a voxel size of 1.65 × 1.65 × 2 mm^3^ (440 × 440 matrices).

Additionally, we simulated the effect of reduced scanning time or injected dose by reconstructing the images from the LAFOV scanner in the time frame of 60 s and 90 s.

Image pre-processing was performed identically to that of the adult cohort used to train the original model [47]. In short, ULDCT images were first resampled to 2 mm³ isotropic voxels, cropped to 512 × 512 to exclude background, and then normalised using a linear scaling of the HU values. The NAC PET images were resampled to the pre-processed ULDCT images and normalised using the 0.5% and 99.5% percentiles.

### 2.3. Model Architecture and Training

In this paper, we utilised a 3D parameter transferred conditional GAN (cGAN) architecture, which is described in detail in our previous work [47]. In short, in the original paper, we proposed to pre-train the generator using a large cohort of 858 patients, followed by training the entire GAN model with a selected subset of the patients for each of the included tracers. In this study, we used an identical architecture and setup with pre-training using the adult cohort but trained the GAN using our paediatric cohort. See the flowchart in Appendix A Figure A1.

The generator consisted of a 3D U-Net with filters [64, 128, 256, 512] that take NAC PET patches of size 128 × 128 × 32 as input and output sCT patches of the same size. The discriminator consisted of a binary classifier with five convolutional layers. It takes paired 3D patches from the NAC PET image and either the real or synthetic CT image as input and outputs a value indicating whether the CT image is real or synthetic.

During each epoch of training, 12 patches were randomly sampled from each subject and subjected to random augmentation (5 degrees rotation, 5 mm translation, and 0.9–1.2 scaling) using TorchIO [50]. The generator was trained using a combination loss that evaluated its performance against the discriminator, the quality of the output images, and a dice loss on the bones. The discriminator was trained on a binary cross-entropy loss. To balance the performance of the pre-trained generator and the discriminator, the discriminator was first trained separately for 50 epochs. Afterward, the generator and discriminator were trained in an adversarial manner for 1450 epochs. The final model was chosen based on visual inspection and the relative mean difference between the generated sCTs and real CTs within organ masks derived from the real CTs from the validation patients.

### 2.4. PET Reconstruction

For PET reconstruction, two reconstructions were performed for each patient in the test set. The sCT was used for attenuation correction to generate the sPET, and the ULDCT was used to generate a standard PET image for reference. Reconstruction parameters remained identical to the NAC PET reconstruction, except for the application of attenuation correction.

The sCTs were generated following the same procedure as described in the original paper. Overlapping NAC PET images were sampled and inputted into the trained generator. The resulting sCT patches were then combined to form the complete sCT image. Finally, the bed from the original CT image was superimposed onto the sCT image.

### 2.5. Data Analysis

#### 2.5.1. Qualitative Evaluation

For qualitative evaluation, an experienced paediatric nuclear medicine specialist with >10 y of experience conducted visual inspection of the PET and sPET images blinded to the underlying AC method by presenting the images side by side in syngo.via (Siemens Healthineers). Artefacts were rated using a Likert scale (0 = none, 1 = minor, 2 = medium, 3 = major). Additionally, any observed differences in image quality were noted, with preference given to the superior image based on a scale of 0–2 (0 = same quality, 1 = insignificant difference, 2 = significant difference).

#### 2.5.2. Quantitative Evaluation

Quantitative evaluation was performed by computing the relative mean difference between the sPET and the reference PET across various organs, utilising organ masks generated by TotalSegmentator [51] from the corresponding ULDCT images. The chosen organs were liver, lungs, kidney, heart, aorta, spleen, brain, bones, colon, oesophagus, and pancreas. Visual inspections were performed to ensure the quality of the segmentation. Furthermore, the nuclear medicine specialist delineated up to five FDG-avid lesions in each PET image individually. The delineation was done using the iso-contouring tool in Mirada (Mirada Medical Ltd., Oxford, UK). We measured the Dice coefficient for delineation overlap and relative percentage difference of SUV mean/max between PET and sPET.

#### 2.5.3. Evaluation of Reduced Counts

To evaluate the robustness of the model towards reduced acquisition times, we used the model trained on full acquisition time data to predict sCT images for the 60 s and 90 s NAC PET images and subsequently reconstructed the sPET, denoted sPET_60_ and sPET_90_, respectively. In addition, anticipating reduced performance on the 60 s data, we trained a second model. The training setup was the same as the default model; however, only trained using the training patients from the Quadra LAFOV scanner. The predicted sCT for the 60 s NAC PET using this low-count model was also used to reconstruct the low-count PET data for the test patients, denoted sPET_60LC_, where LC indicates that the model was fine-tuned on the low-count data. We compared the sPET images to the references PET_60_ and PET_90_ reconstructed using the ULDCT for AC. Evaluation was performed by calculating the relative difference in the organ and lesion masks.

## 3. Results

The visual inspection of the images showed no differences (score 0) between PET and sPET in image quality in 19 out of 23 cases. Four cases showed minor insignificant differences without clinical impact (score 1), primarily related to minor respiration artefacts around the liver or spleen, which were given an artefact score of 1 (minor artefact) by the nuclear medicine specialist. The PET image was deemed the better of the two in three of these cases. No cases showed a significant difference.

Figure 1 illustrates one of the cases where minor insignificant differences without clinical impact between the PET image and the sPET image were found, with the PET image rated as superior. The observed artefact is a respiration artefact around the liver (banana artefact). Despite this artefact, the two PET images appear similar in the figure.

Like the findings in the original paper, the synthetic CT images appear to be free of streaking artefacts caused by metal-induced implants, unlike the CT images. Figure 2 highlights this observation, depicting a patient with an implant in the spine resulting in streaking artefacts evident in the CT but not in the sCT.

In Figure 3, the results of the method on a 1-year-old patient are depicted. It is evident that the produced sPET closely resembles the PET image, although differences are observed in the patient’s left arm, as illustrated in the difference map due to motion between the CT and PET acquisition.

A total of 55 FDG-avid lesions were found across 21 of the 23 examinations. All lesions were found in both PET and sPET, with an average Dice coefficient of 0.92 ± 0.08. The average relative difference was 0.3 ± 4.2% for the mean SUV and 1.4 ± 4.9% for the maximum SUV. The relative difference was within ±10% in nearly all lesions; see Figure 4. Two lesions were in lung tissue (relative mean SUV −3.7% and 4.4%, respectively), and 16 lesions were in bone areas (relative mean SUV 1.6 ± 3.4%, range −6.2% to 5.1%).

The quantitative ROI evaluation also showed a mean SUV relative difference of −2.6 ± 5.8% across all organs (Figure 5), with the lung and colon being the organs with the largest difference between PET and sPET, which is likely due to the respiratory motion between PET and CT, whereas sPET is purely based on the PET signal.

The low-count images sPET_60_ and sPET_90_ had an overall average relative PET difference within the selected organs that was comparable to the results obtained with the full-count sPET images, albeit with increased standard deviation, which was worst for the sPET_60_ image. The fine-tuned model (sPET_60LC_) resulted in identical quantitative results as the full-count model (Figure 6).

## 4. Discussion

In this study, we extended a previously introduced deep learning method for generating synthetic CT images from PET emission data for use in paediatric patients. Our primary goal was to minimise the radiation exposure to paediatric patients during PET/CT scans. Furthermore, by directly deriving the attenuation map from the PET data, we ensure excellent alignment between the PET and attenuation images, thus addressing concerns regarding motion-induced artefacts.

The quantitative analysis revealed a mean SUV relative difference of −2.6 ± 5.8% across all organs (Figure 5), with the largest discrepancies occurring in the lung and colon, likely due to respiratory motion. This overall error rate is comparable to the error reported in similar studies by Dong et al. and Armanious et al. of 0.1% and −0.8%, respectively [43,44]. Direct comparison between the results is challenged by a difference in the cohort, as both of these works include only adult patients. The observed discrepancies around the lung and diaphragm highlight the potential of this method to reduce motion-related artefacts, including gross motion, as illustrated in Figure 3, consistent with findings from previous studies. This capability is particularly important in paediatric imaging, as children are more susceptible to motion artefacts due to difficulty in remaining still and the more frequent need for sedation. Mitigating these artefacts effectively is, therefore, crucial for enhancing diagnostic accuracy in this patient population, making this method a valuable alternative to other radiation-lowering techniques such as ULDCTs.

Furthermore, 55 FDG-avid lesions were identified across 21 of the 23 examinations, detected in all cases in both PET and sPET images. We achieved a high agreement both regarding delineation (average Dice coefficient of 0.92 ± 0.08) and quantitatively (relative mean SUV 0.3 ± 4.2%, relative max SUV 1.4 ± 4.9%). These results align with those of Dong et al. [43], who reported a mean difference of 1.1% in lesions, and Armanious et al. [44], who noted a mean deviation of 0.9%. We did not observe any decrease in performance for the lesions that were located within bone or lung tissue, areas that are otherwise known to be especially challenging for NAC-PET-based methods [39], mainly attributed to an overall low PET uptake and high tissue heterogeneity.

Qualitative results revealed no differences in image quality between PET and sPET in 19 out of 23 cases. In the remaining four cases, minor insignificant differences with no clinical impact were observed, primarily attributed to minor respiration artefacts around the liver or spleen. No cases exhibited significant differences. Moreover, the method successfully eliminated streaking artefacts caused by metal implants in the observed test patients and yielded promising results for patients as young as 1-year-old. These results indicate that the proposed method is ready for testing in clinical routine.

Finally, since the LAFOV scanners allow for short acquisition times, we also evaluated the generalisability of our model in such a low-count setting. Here, we demonstrated retained quantitative performance as low as 60 s PET acquisition, albeit with the best performance when fine-tuning the model with the low-count data. These results motivate the use of our model in such fast acquisition setups.

This study has certain limitations. Firstly, due to the limited number of very young patients, this demographic of patients has been included to a limited degree in the training and evaluation of the model. As a result, the performance of the model on very young or small patients remains somewhat uncertain. Secondly, the model was trained using ULDCTs, meaning it was designed to output ultra-low-dose CT images, which may be of lower quality compared to traditional low-dose or standard-dose ACCT images.

Despite these limitations, the model demonstrated promising results, both in its potential to generate attenuation-corrected PET images without the need for ACCT, thereby reducing overall radiation exposure, as well as in its ability to mitigate motion artefacts. In the future, we plan to expand the implementation of the model to include more patients, which will provide a better understanding of its performance on very young patients. Additionally, we aim to extend the approach for use with additional radiopharmaceuticals used to examine the paediatric cohort.

## 5. Conclusions

In this study, we extended our deep learning method to generate synthetic CT images from non-attenuation correction emission data for paediatric patients, successfully reducing radiation exposure by eliminating the need for a CT scan. The method showed excellent qualitative and quantitative performance and effectively mitigated motion-induced artefacts. High agreement in lesion detection between PET and sPET images supports its clinical applicability. Despite limitations, such as the limited inclusion of very young patients, the model demonstrated robust performance even in low-count settings. Our findings suggest this approach is a valuable alternative to traditional methods, ready for further clinical testing to enhance diagnostic accuracy and patient safety in the younger population and reduce the occurrence of late sequelae in children.

## Figures and Tables

**Figure 1 diagnostics-14-02788-f001:**

Illustrative sample patient with banana artifact presented. Panels (**a**,**b**) show the normal CT and corresponding PET. The synthetic CT (sCT) and corresponding sPET are seen in (**c**,**d**). PET is fused on top of the CT scan in (**e**), illustrating the mismatch between CT and emission data. The blue line represents the superior part of the liver at the time of CT scanning. Panel (**f**) shows the sPET fused on top of the sCT.

**Figure 2 diagnostics-14-02788-f002:**
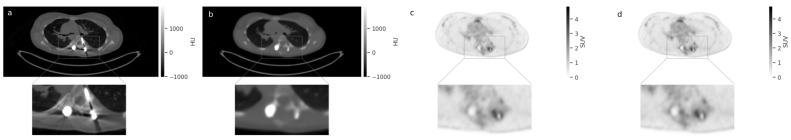
Sample patient with metal implant exhibiting streaking artefacts in the CT image (**a**), which are absent in the sCT image (**b**). The corresponding PET images are seen for PET and sPET, respectively (**c**,**d**). The zoom panels have been magnified by a factor of 2.3.

**Figure 3 diagnostics-14-02788-f003:**
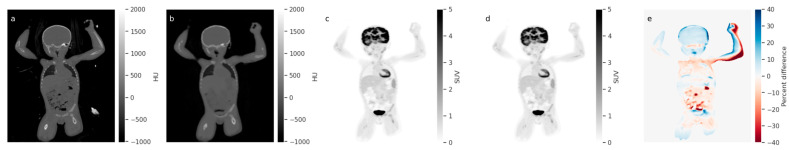
Representation of a 1-year-old patient featuring the CT, sCT, and corresponding PET images PET and sPET (**a**–**d**). Additionally, a relative percent difference map between the PET and sPET images (**e**) highlights that the discrepancies in the PET images are localised in the patient’s cranium and left arm.

**Figure 4 diagnostics-14-02788-f004:**
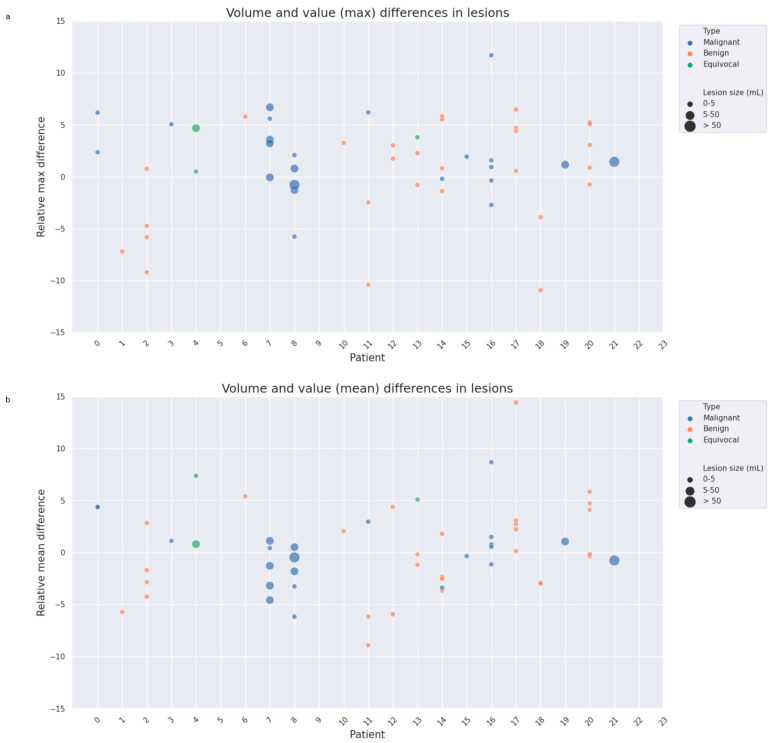
Relative max difference (**a**) and mean difference (**b**) between the PET and sPET for lesions found in the examinations. The colour and size of each point represent the lesion type and size, respectively.

**Figure 5 diagnostics-14-02788-f005:**
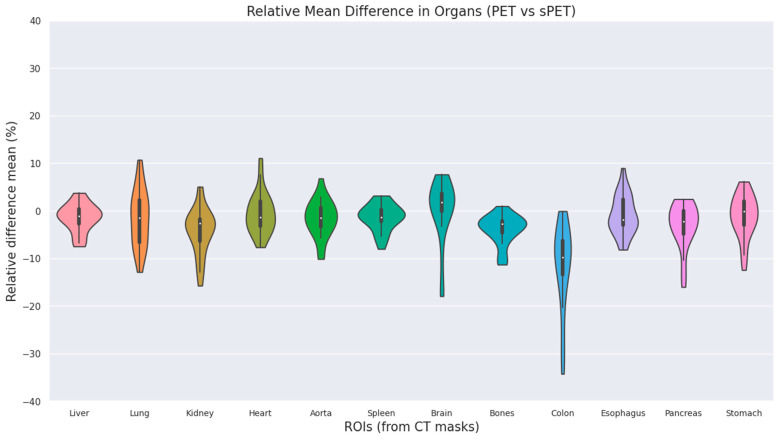
Violin-plot showing the mean relative percent difference between PET and sPET for selected organs. The white dot in each presents the median value, and the solid black box represents the interquartile range, whereas the line extends to 1.5 times the interquartile range.

**Figure 6 diagnostics-14-02788-f006:**
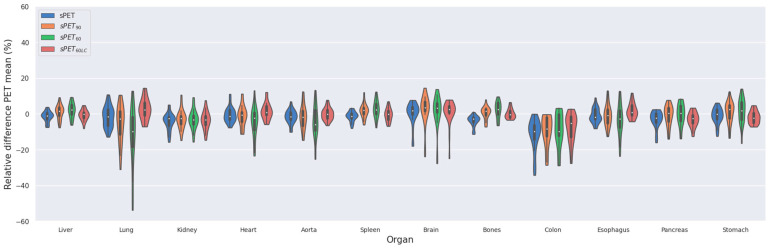
Violin-plot showing the mean relative difference between PET, PET_60_, sPET_90_, and sPET_60LC_ for selected organs.

**Table 1 diagnostics-14-02788-t001:** The three cohorts used in this study from the Siemens Biograph Vision 600 and LAFOV Siemens Biograph Vision Quadra scanner, Siemens Healthineers, Knoxville, TN, USA.

Cohort	Scanner	Inclusion Period	*n* Total Examinations [M/F]	*n* ≤ 6 Years	Weight (kg)	Age (Years)
Train	Siemens Biograph Vision 600 PET/CT	August 2021 to March 2022	81 (45/36)	21/81	8.5–78	0.7–19
Train	LAFOV Siemens Biograph Vision Quadra PET/CT	November 2021 to June 2023	91 (48/43)	23/91	4–92	0–18
Test	LAFOV Siemens Biograph Vision Quadra PET/CT	May 2022 to June 2023	23 (9/14)	3/23	13–94	1–18

## Data Availability

Data supporting reported results can be obtained via contact with the corresponding author upon reasonable request and legal approval. The data are not publicly available due to no public data sharing agreement.

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
