# Peer review of "CT-Free Attenuation Correction in Paediatric Long Axial Field-of-View Positron Emission Tomography Using Synthetic CT from Emission Data"

_diagnostics, 2024, doi:10.3390/diagnostics14242788_

Round 1

Reviewer 1 Report

Comments and Suggestions for Authors

This manuscript aims to reduce the dose in pediatric PET/CT examinations and can be applied in clinical practice to ensure patient radiation dose of safety.
Please refer to the comments below when writing your manuscript.

1. Line 169. Is this the official title of nuclear medicine specialist? Or are you a nuclear radiologist?

2. In patient cohort. Data from pediatric patients under 6 years of age were used. It seems necessary to create a table and explain whether weight, age, etc. were classified.

3. In results. In conclusion, it was said that radiation expoure reduction was successful. However, no specific details on dose reduction can be found in the results. Please quantitatively explain the dose reduction in the results.

Thank you.

Author Response

Comments 1: This manuscript aims to reduce the dose in pediatric PET/CT examinations and can be applied in clinical practice to ensure patient radiation dose of safety. Please refer to the comments below when writing your manuscript.
Response 1: We thank the reviewer for taking the time to review this manuscript. Please find the detailed responses below and the corresponding revisions/corrections highlighted/in track changes in the re-submitted files.

Comments 2: Line 169. Is this the official title of nuclear medicine specialist? Or are you a nuclear radiologist?

Response 2: The term nuclear medicine specialist is an official term. This term reflects the expertise and qualifications of medical doctors who specialize in nuclear medicine, encompassing diagnostic applications of radiopharmaceuticals.

Comments 3. In patient cohort. Data from pediatric patients under 6 years of age were used. It seems necessary to create a table and explain whether weight, age, etc. were classified.

Response 3: We agree this information is important to add. We have extended Table 1 with two extra columns: Weight and Age, each displaying the range in kg and years, respectively.

Comments 4: In results. In conclusion, it was said that radiation expoure reduction was successful. However, no specific details on dose reduction can be found in the results. Please quantitatively explain the dose reduction in the results.

Response 4: The dose reduction is reached by avoiding acquiring a CT image for attenuation correction. We have clarified the sentence in the conclusion to: “[…] successfully reducing radiation exposure by eliminating the need for a CT scan."

Reviewer 2 Report

Comments and Suggestions for Authors

This study aims to minimize radiation exposure by generating synthetic CT (sCT) images from emission PET data, eliminating the need for attenuation correction (AC) CT scans in paediatric patients.

The introduction gives a sufficient overview about background and the goal of the study. 

The methods are well described, nevertheless a simple flow diagram would facilitate the understanding of the workflow of methodss the authors did apply.

Attenuation correction is indispensable for avoiding both qualitative and quantitative PET errors which could compromise diagnostic accuracy. How about the precision inside an organ, presented are more or less total body images. Please comment on this.

With your approach, the non-attenuation corrected (NAC) PET can be used to synthesise the attenuation and scatter corrected PET image directly, or to synthesise the AC CT image. Previous work showed that the errors were large in the lung, mainly owing to tissue heterogeneity, and almost no evaluation results on bone lesions were reported. Please discuss this point in the light of your own findings. It should be noted that the density errors in deep learning approach directly leads to PET quantification error in the image space. How can you avoid or correct this?

Author Response

Comments 1: This study aims to minimize radiation exposure by generating synthetic CT (sCT) images from emission PET data, eliminating the need for attenuation correction (AC) CT scans in paediatric patients. The introduction gives a sufficient overview about background and the goal of the study.

Response 1: We thank the reviewer for taking the time to review this manuscript. Please find the detailed responses below and the corresponding revisions/corrections highlighted/in track changes in the re-submitted files.

Comments 2: The methods are well described, nevertheless a simple flow diagram would facilitate the understanding of the workflow of methodss the authors did apply.

Response 2: We have added a flowchart describing the training setup including pretraining from the adult cohort. The figure is added as Supplementary Figure A1 and a reference was added to the methods section.

Comments 3: Attenuation correction is indispensable for avoiding both qualitative and quantitative PET errors which could compromise diagnostic accuracy. How about the precision inside an organ, presented are more or less total body images. Please comment on this.

Response 3: We agree that it is important to validate an attenuation correction method locally as global metrics might hide local errors, and we believe we have done so in the quantitative evaluation. We have analysed the mean relative difference between PET and sPET within 12 regions of interests (11 organs and a bone mask). Here we found excellent overall agreement and reported the range for single patients in Figure 5. Moreover, we performed a mean and max analysis of up to five FDG-avid lesions for each patient (Figure 4), which furthermore gives an indication of the regional performance, as these lesions can be distributed throughout the body.

Comments 4: With your approach, the non-attenuation corrected (NAC) PET can be used to synthesise the attenuation and scatter corrected PET image directly, or to synthesise the AC CT image. Previous work showed that the errors were large in the lung, mainly owing to tissue heterogeneity, and almost no evaluation results on bone lesions were reported. Please discuss this point in the light of your own findings. It should be noted that the density errors in deep learning approach directly leads to PET quantification error in the image space. How can you avoid or correct this?

Response 4: Thank you for pointing these areas out. We have added to the lesion analysis the results for bone and lung lesions that demonstrates the same performance for these lesions compared to all lesions. We have added a paragraph to the results: “Two lesions were in lung tissue (relative mean SUV -3.7% and 4.4%, respectively) and 16 lesions were in bone areas (relative mean SUV 1.6±3.4%, range -6.2% to 5.1%).”.

We have also added the following sentence to the discussion: “We did not observe any decrease in performance for the lesions that were located within bone or lung tissue, areas that are otherwise known to be especially challenging for NAC-PET-based methods [39], mainly attributed to an overall low PET uptake and high tissue heterogeneity.”